# The Effects of the ACTIVE VALUES Program on Psychosocial Aspects and Executive Functions

**DOI:** 10.3390/ijerph20010595

**Published:** 2022-12-29

**Authors:** José Francisco Jiménez-Parra, Noelia Belando-Pedreño, Alfonso Valero-Valenzuela

**Affiliations:** 1Faculty of Sport Sciences, Department of Physical Activity and Sport, University of Murcia, 30720 Murcia, Spain; 2Faculty of Sports Sciences, Department of Physical Activity and Sport, European University of Madrid, 28670 Madrid, Spain

**Keywords:** personal and social responsibility, active breaks, physically active learning, motivation, cognitive performance, physical education

## Abstract

The main objective of this study was to implement an educational program named ACTIVE VALUES and to analyse the psychosocial and cognitive effects of its application. It is a quasi-experimental repeated measures research with a non-randomised experimental group (EG) and a control group (CG). The sample consisted of 102 students in the 6th grade of primary school, aged between 11 and 13 years (*M* = 11.59; *SD* = 0.60), and 4 teachers aged between 27 and 52 years (*M* = 38.5). The intervention program lasted 4 months, in which the EG implemented a teaching methodology based on the incorporation of classroom-based physical activity (CB-PA) in the structure of the Teaching for Personal and Social Responsibility (TPSR) model to develop personal and social values in students, as well as to reduce children’s sedentary behaviour in the classroom in different educational areas (e.g., mathematics, Spanish language, social sciences and natural sciences), while the CG used a conventional methodology based on direct instruction. The main results found show significant improvements in intrinsic motivation variables (including intrinsic motivation for achievement, stimulating experiences and knowledge), self-determination index, autonomy, relatedness, psychological mediators index, personal and social responsibility, teacher climate, intention to be physically active and executive functions in the EG, while amotivation values increased in the CG. In conclusion, interdisciplinary educational programs based on the combination of pedagogical models and active methodologies are postulated as methodological alternatives to achieve an integral and multilateral development of children and adolescents, as well as to improve the different learning domains of physical education, such as cognitive, social and motor. It is recommended that future research should consider longitudinal designs with mixed methods and follow-up data to assess learning retention, as well as larger samples and the measurement of a greater number of executive functions (e.g., inhibitory control and attention).

## 1. Introduction

The promotion of healthy lifestyle habits in the school population, for the maintenance of integral health throughout the maturational development and for its later adult life, could be one of the great challenges of the educational system in coordination with the health system [1,2,3]. One of the main causes of this great social, educational and health commitment is the high rate of childhood obesity, a multisystem pathology that has reached the epidemic level worldwide [4]. In addition, in this multifactorial health approach, attention must be paid to the behavioural level of the youngest, referring to behaviours of a psychosocial nature, such as personal and social responsibility, self-management of emotions, empathy and social bonding, among others [5]. It has been shown that physical activity (PA) in the school is associated with physical and psychological well-being, and academic and cognitive performance [6,7], positively influencing different processes of cognition in general [8] and executive functions in particular [9]. However, recent studies state that school-aged youth groups do not meet PA recommendations and are increasingly sedentary [10,11,12].

With the aim of providing greater opportunities to increase PA in the classroom, a strategy known as classroom-based physical activity (CB-PA) [13] is presented as an intervention proposal that can help prevent these problems and improve healthy habits. CB-PA is, specifically, PA performed during school time, and not during the school break or lunch period [13]. It is classified into 2 types or methods: (a) active breaks, consisting of short periods (between 10 and 15 min) of moderate to vigorous intensity physical activity (MVPA) that are integrated into the curricular routine, that are implemented by the teacher during or between academic instruction and do not require special material for their development; and (b) physically active learning, consisting of teaching curricular content (high cognitive involvement) through games or PA tasks so that students learn new content or retain learning (curriculum-focused active breaks) [13]. Studies related to CB-PA have reported benefits in cognitive [14,15,16] and academic performance [17,18,19], and students’ health [20,21,22] associated with increasing PA and reducing sedentary behaviours [23,24]. CB-PA has a positive influence on the children’s cognitive processes [25,26] and, more precisely, on the performance of executive functions [27,28], young students’ life satisfaction and well-being [29,30] and increase motor participation and interaction [31].

The educational context is one of the ideal contexts for the development of values, competences and executive skills that improve personal and social well-being, increasing students’ abilities to face the situations of daily life [32]. With the aim of providing methodological coherence to the implementation of these educational programs, emerging and consolidated pedagogical models are used, in addition to the hybridization of models [33,34,35], which ensure good practice in terms of teaching strategies, procedures and methods of assessment that are faithful to the internal structure of these models.

In this regard, the Teaching Personal and Social Responsibility (TPSR) model, developed by Donald Hellison [5] with the purpose of enhancing personal and social capacities in children and adolescents at risk of social exclusion [36], has become one of the models most implemented in the educational context [37]. This consolidated pedagogical model is structured into five levels of work on intrapersonal values, such as personal responsibility, autonomy and self-management, and interpersonal values such as social responsibility, coexistence and group cohesion. The five aspects to be dealt with in a typical educational lesson adapted to the curricular contents of a subject are: (1) respect for the rights and feelings of others, (2) participation and involvement, (3) self-management and autonomy, (4) help and leadership and (5) transfer outside the classroom (the objective is that all the values learned are transferred into society and in their day-to-day lives beyond the classroom).

The model allows teachers to include innovative intervention strategies, in which the student is the protagonist of the teaching–learning process. Some examples of these strategies are: during class, the teacher attends to all students, especially those with the greatest difficulties; one of several students are encouraged to explain to the rest of the class what the activity consists of, to which they have to respond as a group and by consensus, and at the end of the session there is a reflection (teacher–student). In this regard, numerous studies, such as those carried out by Hellison and Walsh [38], Hellison and Martinek [39] or Li et al. [40] link high levels of personal and social responsibility with states of intrinsic motivation in physical education.

These behavioural consequences show that motivation is closely linked to the teaching–learning process and is explained under a consolidated theoretical framework such as the self-determination theory (SDT) [41]. The theory focuses on the socio-contextual factors that promote or hinder the progress of students through the satisfaction of their basic psychological needs (BPN) [42]. When the pedagogical designs implemented by teachers satisfy these needs for autonomy, competence and relatedness, students are more likely to present a more self-determined degree of motivation to participate in learning tasks [43,44] and present more physically active behaviours [45].

Therefore, the aim of this research was to implement an educational program named ACTIVE VALUES and to analyse the psychosocial and cognitive effects of its application. It is hypothesized that the development of personal and social responsibility strategies together with the promotion of healthy PA will have positive behavioural consequences such as the development of executive functions, the improvement of personal and social responsibility and social school climate, and an increase in levels of self-determined motivation and intention to be physically active outside of the classroom.

## 2. Materials and Methods

### 2.1. Study Design

The present study followed a quantitative methodology (tests and questionnaires) and a quasi-experimental pre-test/post-test research design (repeated measures), with a non-randomised experimental group (EG) and control group (CG).

### 2.2. Participants

The study sample included two primary schools located in two municipalities of the Region of Murcia (Spain) with a medium-low socioeconomic level. A total of 4 teachers and 102 students in 6th grade of primary education, aged between 11 and 13 years (*M* = 11.59; *SD* = 0.60), were selected by convenience and accessibility and divided into 2 groups:

(1)EG: composed of 2 teacher–tutors, aged 27 and 39, with an average teaching experience of 8 years. This group included 49 students with an average age of 11.57 years.(2)CG: composed of 2 teacher–tutors, aged 36 and 52, with an average teaching experience of 18 years. This group included 53 students with an average age of 11.60 years.

Inclusion criteria for participation in this study were: (a) completion of all pre- and post-intervention data collection tests, answering at least 90% of the items; and (b) attendance at 80% or more of the classes.

### 2.3. Instruments

#### 2.3.1. Executive Functions

Executive functions were assessed using two tests from the NIH Examiner program [46]: (a) *verbal fluency* by categories, in which students had to write the maximum number of words related to a category (e.g., animals or vegetables) in 60 s; and (b) unstructured task (*planning*), in which students had to solve the maximum number of brainteasers in a maximum time of 6 min. Participants had to complete four puzzle pages (four games per page) to reach the highest number of points possible, as each game had a certain score depending on the difficulty.

#### 2.3.2. Psychosocial Variables

Psychosocial variables were collected by means of a closed-ended questionnaire consisting of two parts: (a) sociodemographic questions and (b) questionnaire scales with validity and reliability for the study population. These scales were:

(1)*Academic motivation scale*: The Spanish validation [47] of the Échelle de Motivation en Éducation (EME) [48] was used, consisting of 28 items divided into 7 subscales: intrinsic motivation (IM) towards knowledge, IM towards achievement, IM towards stimulating experiences, external regulation, introjected regulation, identified regulation and amotivation. Students responded on a seven-point Likert scale from one (not at all matched) to seven (fully matched). Pre- and post-test Cronbach’s alpha values were: amotivation (0.900 and 0.911), external regulation (0.853 and 0.876), introjected regulation (0.873 and 0.862), identified regulation (0.854 and 0.817), IM towards knowledge (0.809 and 0.814), IM towards achievement (0.878 and 0.835), IM towards stimulating experiences (0.858 and 0.819) and IM (0.846 and 0.863). These variables were unified to calculate the self-determination index (SDI), a valid and reliable measure for this population [49], using the following formula: ((intrinsic motivation * 2 + identified regulation) − (introjected regulation + external regulation)/2 − (amotivation * 2)) [50].(2)*Personal and social responsibility scale:* The Personal and Social Responsibility Questionnaire (PSRQ), adapted to the school context [40] and to Spanish [51], was used to measure responsibility. The PSRQ is composed of 14 items grouped into 4 levels (respect for others, participation and effort, autonomy and help, and leadership) and 2 dimensions: social responsibility (7 items) and personal responsibility (7 items). Students answered the questionnaire on a Likert-type scale from one (strongly disagree) to six (strongly agree). The reliability value in the pre-test for social responsibility was 0.837 and 0.871 for personal responsibility, while in the post-test it was 0.839 and 0.852, respectively.(3)*Basic Psychological Needs Scale:* The Psychological Need Satisfaction in Exercise Scale (PNSE) designed by Vlachopoulos and Michailidou [52] was used to measure basic psychological needs; this scale has been validated in the Spanish educational context of secondary education [53] and primary education [54]. The PNSE is composed of 12 items divided into 3 dimensions with 4 items each: (1) autonomy, (2) competence and (3) relatedness. Students responded on a Likert-type scale from one (strongly disagree) to five (strongly agree). Pre-test reliability was 0.849 for autonomy, 0.842 for competence and 0.868 for relatedness to others, while post-test reliability was 0.839, 0.846 and 0.864, respectively. The psychological mediators index (PMI), a valid and reliable measure [55], was also applied to assess the set of the 3 basic psychological needs, obtaining an internal consistency in the pre-test of 0.880 and in the post-test of 0.906.(4)*School Social Climate Scale:* the questionnaire to assess school social climate (CECSCE) was used; this test was developed and validated by Trianes et al. [56] in the Spanish educational context, based on the California School Climate and Safety Survey [57]. This scale is composed of 14 items grouped into 2 dimensions or factors: (1) school climate (8 items) and (2) teacher climate (6 items). Participants answered the scale on a Likert-type scale with a minimum value of one (strongly disagree) and a maximum of five (strongly agree). Pre-test reliability was 0.826 for school climate and 0.831 for teacher climate, while post-test reliability was 0.833 and 0.843, respectively.(5)*Intention to be Physically Active Scale (IPAS):* The Intention to be Physically Active Scale (IPAS) by Hein et al. [58] was used to measure the intention to be physically active,; this scale has been validated in the Spanish school context of secondary education [59] and primary education [60]. The scale is composed of five items with a single dimension. Students responded on a Likert-type scale ranging from one (strongly disagree) to five (strongly agree). The pre-test reliability was 0.913 and the post-test reliability was 0.921.

### 2.4. Procedure

First, the study protocol was designed, which has recently been published [61], and approval was requested from the Ethics Committee of the University of Murcia. Secondly, after obtaining approval (3207/2021), a letter of introduction to the schools to participate in the study was drafted and sent. Thirdly, informed consent was requested from the students and their parents (treatment of confidential data and participation in the study), as well as from the teachers (recording of the sessions), following the ethical guidelines of the American Psychological Association. Fourthly, data collection (pre-test) was carried out in the classroom in an environment isolated from distractions. The tests were administered according to the following distribution to ensure maximum student concentration: (1) tests of executive functions (20 min), (2) 15 min break and (3) psychosocial aspects questionnaire (30 min). Before starting the collection, a presentation was given to the students on how to complete the tests and the questionnaire to ensure their understanding. They were told that the answers were anonymous and would not affect their grade, but that they should answer honestly. Both the teacher and the principal investigator were present in the classroom to answer any questions. The same procedure was followed for the post-test.

#### 2.4.1. Teacher Training

The implementation of any teaching methodology or program requires specific professional development for teachers [62]. To this end, the present research followed a 2-phase approach: (1) *Initial Training*, in which a 15 h theoretical–practical course was held in which the teacher was explained the theoretical foundations and the global and specific strategies (e.g., session structure, levels of personal and social responsibility, conflict resolution strategies, etc.) for developing the TPSR (10 h), as well as the methods for incorporating CB-PA (active breaks and physically active learning) and combining it with TPSR (5 h); (2) *Continuous Professional Development*, in which follow-up strategies were carried out every 3–4 weeks, such as training seminars, consultation and feedback through observational analysis of sessions and report writing to provide guidance and support during the implementation of the program [63].

#### 2.4.2. ACTIVE VALUES Program

An interdisciplinary educational program, named ACTIVE VALUES, based on the integration of CB-PA into the structural elements of the TPSR model, was implemented in primary education classrooms. Teachers taught the contents of the subjects of mathematics, Spanish language and natural and social sciences, as set out in the Spanish education law [64], for 4 months in 60–90 min sessions (20 h per week).

The ACTIVE VALUES program developed the essential elements of the TPSR proposed by Hellison [2] in a flexible way [65]. The responsibility levels of respect for others (level 1), participation and effort (level 2), autonomy (level 3) and helping others and leadership (level 4) were worked on through interconnection and progression, presenting a level every 4 weeks. Transfer (level 5) was the only one that was developed together with the other levels to increase its relevance in the students’ daily lives, as this is an essential component of the TPSR model [66,67].

Teachers incorporated the CB-PA into the TPSR session structure [5] adapted to general education [68], which has the following parts: (1) awareness raising, (2) active responsability, (3) group meeting and (4) self- and co-assessment. During the active responsibility phase (part 2), teachers proposed CB-PA that were related to motor development (active breaks), the elements of responsibility and to the curriculum content (physically active learning). The methods used to implement the CB-PA were those proposed in previous research [31,35,61].

#### 2.4.3. Intervention Fidelity

Following the aspects to be taken into account to ensure fidelity in research based on the application of pedagogical models proposed by Hastie and Casey [69], the Tool for Assessing Responsibility-based Education and Active Breaks (TARE-AB) in the classroom was used to validate the implementation of the ACTIVE VALUES program. This instrument is a combination of the Tool for Assessing Responsibility based Education (TARE 2.0) [67,70] and the Tool for Assessing Active Breaks (TAAB) [31]. In addition, teachers were invited to evaluate their own performance at the end of each session to reflect on the implementation of the ACTIVE VALUES program [61]. A SONY HDR video camera was used to record two sessions every four weeks. The camera was placed prior to the intervention to promote spontaneous behaviours among teachers and students [71].

An observational analysis was carried out by two graduates in PA and sport sciences (observers) with knowledge in educational research based on active methodologies, who were specifically trained through a 10 h course based on the use of TARE-AB and taught by expert researchers on CB-PA and TPSR, following the sequence proposed by Wright and Craig [67]. At the end of the course, inter- and intra-reliability was calculated; this was above 80% and guaranteed the necessary reliability to start the analysis of the study sessions. Total agreement (TA) was calculated using the following formula: total number of agreements (TA) divided by agreements (A) plus disagreements (D) (TA = TA/A + D) [72].

### 2.5. Data Analysis

Data analysis was performed with IBM SPSS 22.0. The internal consistency and reliability of all scales used in the present study were assessed by conducting a preliminary analysis. Based on 102 cases, McDonald’s omega [73] on the 5 scales ranged from 0.683 to 0.932, values considered acceptable by authors such as Sturmey et al. [74]. Furthermore, this coefficient presents a more feasible value for social science studies with non-continuous variables, although its use is more appropriate in five-point Likert-type scales [75]. However, in this study, two scales with more than five points were included; their values were lower than the five-point scales. The omega coefficient is not affected by the sample error or the number of items, among other issues.

On the other hand, the Kolmogorov–Smirnov test (*p* > 0.05) was performed to test the hypothesis of normality. The Kolmogorov–Smirnov test for normality rejected the hypothesis of normality for most variables, except for intrinsic motivation, personal responsibility and PMI, so non-parametric tests were used to analyse inferences. After that, the Mann–Whitney *U* test was used to compare the variables between EG and CG. This comparison was performed both before and after the intervention. Finally, to compare intra-group variables before and after the intervention, the Wilcoxon signed-rank test was used.

## 3. Results

Table 1 shows the means and standard deviations of all variables differentiated by groups (CG and EG) and time of intervention (pre/post), including the *p*-values obtained with the non-parametric comparative statistical tests.

The Mann–Whitney *U* test used to compare the variables between the two groups (control and experimental) before the intervention revealed no significant differences. This aspect shows the homogeneity existing between both groups, with respect to the variables under study, before starting the implementation of the interdisciplinary educational program.

The results of the Wilcoxon signed-rank test show that CG only slightly improved in social responsibility (*p* = 0.047) over the course of the study, while it significantly decreased in intention to be physically active (*p* = 0.014). EG significantly increased the levels of most variables throughout the process, except for external regulation (*p* = 0.242), introjected regulation (*p* = 0.498), competence (*p* = 0.252) and school climate (*p* = 0.643). On the other hand, EG amotivation decreased significantly (*p* = 0.004) after the intervention.

Regarding intergroup differences post-test, the Mann–Whitney *U* test revealed that there were statistically significant differences in favour of the EG on the variables of IM of achievement (*p* = 0.019), IM of stimulating experiences (*p* = 0.020), IM of knowledge (*p* = 0.006), overall IM (*p* = 0.001), SDI (*p* = 0.001), autonomy (*p* = 0.000), relatedness (*p* = 0.010), PMI (*p* = 0.000), personal responsibility (*p* = 0.005), social responsibility (*p* = 0.000), teacher climate (*p* = 0.049), intention to be physically active (*p* = 0.000) and planning (*p* = 0.013). The variable amotivation showed significant differences in favour of CG (*p* = 0.005).

## 4. Discussion

In order to deepen our knowledge of the effects of the ACTIVE VALUES educational program based on the combination of the TPSR model and CB-PA on psychosocial and cognitive variables, the hypothesis on executive functions was partially confirmed. After the intervention, the experimental group presented statistically significant improvements in the executive development of planning but not of fluency. In the study carried out by Muñoz-Parreño et al. [76], where these 2 variables were measured together with working memory, inhibition and cognitive flexibility in primary school students of similar ages to whom a program based on CB-PA was applied, all these functions improved compared to the control group that received their curricular areas without any type of PA for 17 school weeks. For their part, Vazou and Smiley-Oyen [77], who applied a program of active breaks during mathematics classes to a group of primary school children, found that inhibitory control improved, but no improvements were obtained in selective attention or in working memory. Finally, Greeff et al. [8], in their meta-analysis study of the effects of PA on executive functions, concluded that positive effects were found on the improvement of attention and academic performance, and that it was greater the more continuous the PA was throughout the course of time of the different weeks. This leads to the belief that the CB-PA programs cause improvements in cognitive functions, but these vary depending on the study.

In addition, in the study carried out here, a greater intention to be physically active outside school hours was found, which may be due, at least in part, to the promotion of PA that has been carried out in classrooms with an increase in the levels of PA in children, as was the case in the studies by Contento et al. [78], Schwazer et al. [79], Dunton et al. [80] and Greeff et al. [81] with students aged around the last year of primary school and the beginning of secondary education. However, other studies have not reported increases in PA levels after a program based on active breaks, as was the case by Whitemore et al. [82], Tymms et al. [83] or Spruijt-Metz et al. [84], which can be explained by a potential compensation mechanism, whereby children exert lower levels of activity after active lessons to compensate for their earlier increased exertion [18]. In any case, these improvements in the intention to continue being physically active are also in line with the results obtained in other studies that have implemented the TPSR program without combining it with other methodologies [85].

In terms of psychological variables, the EG experienced significant changes in the values of personal and social responsibility, teacher climate, the satisfaction of basic psychological needs and the degree of academic motivation. In addition, a decrease in the levels of amotivation was observed after the intervention. Accordingly, other studies that have hybridized the TPSR with sport education have shown improvements in the satisfaction of basic psychological needs (competence and relatedness), in enjoyment [86] and in social development [87]. The study carried out by García-Castejón et al. [88], where the TPSR was hybridized with teaching games for understanding, also showed improvements in autonomous motivation, in psychological mediators and in enjoyment, in addition to gains in personal and social responsibility. However, in the investigations that have implemented CB-PA and have measured variables related to well-being and satisfaction with life, although most report benefits in enjoyment, there are few positive findings in well-being [29] and academic motivation [89]. Therefore, it is highly recommended to carry out more studies in this line that can consolidate the positive results obtained in this investigation and discern whether it is due more to CB-PA or to the TPSR program.

The limitations of the study include the type of research design, quasi-experimental, as well as the lack of randomisation of intervention groups. An experimental design could have established causal relationships between the application of the intervention and the effects on the variables under study. Another limitation is the absence of a follow-up plan to analyse the evolution of active behaviour outside school hours once the CB-PAs were discontinued. Studies analysing the effects of CB-PA outside school hours on PA behaviours are needed to significantly contribute to the increase in levels of MVPA [90,91]. Finally, the analysis of two executive functions is limited for understanding and extending the causal effects of the program on this variable.

Prospective research should focus on the development of experimental studies in which educational programs are implemented with a process of intervention reliability [92] to develop young students’ awareness of their personal and social responsibility, increase autonomous behaviours in their educational and daily activities, stimulate executive skills and increase levels of self-determined motivation and a more active lifestyle, both inside and outside of school hours. Longitudinal interventions during school hours and after school are needed, aimed at improving physical performance and based on cognitive and psychosocial factors such as executive functions, motivational regulation and physical self-concept. In addition, future studies should measure additional executive functions, such as inhibitory control or cognitive flexibility, to better understand the intervention effects of the program on cognitive performance.

## 5. Conclusions

Significant positive effects of the ACTIVE VALUES educational program were observed on executive functions, personal and social responsibility, teacher climate, satisfaction of BPN and academic motivation in schoolchildren. Likewise, the program based on the model of personal and social responsibility and CB-PA contributed to decreasing the levels of amotivation in the EG and improved the intention to be physically active.

Interdisciplinary educational programs based on the combination of pedagogical models and active methodologies are postulated as methodological alternatives to achieve an integral and multilateral development of children and adolescents, as well as to improve the different learning domains of physical education: cognitive, social, affective and motor. It is recommended that future research consider longitudinal designs with mixed methods and follow-up data on learning retention, as well as larger samples and the measurement of a greater number of executive functions (e.g., inhibitory control and attention) and psychological variables, such as physical self-concept.

## Figures and Tables

**Table 1 ijerph-20-00595-t001:** Pre-and post-test differences in the subscales according to the group.

		Pre-Test					Post-Test					Pre/Post Intra-Group Test Differences	
		Control		Experimental			Control		Experimental			Control	Experimental
Variables	Factors/ Dimensions	Mean	*SD*	Mean	*SD*	Mann–Whitney *p*-Value	Mean	*SD*	Mean	*SD*	Mann–Whitney *p*-Value	Wilcoxon *p*-Value	Wilcoxon *p*-Value
Academic Motivation	Amotivation	1.53	0.75	1.58	0.71	0.637	1.60	0.75	1.22	0.44	0.005 **	0.102	0.004 **
	External R.	6.10	0.64	6.08	0.58	0.917	6.11	0.61	5.96	0.51	0.125	0.841	0.242
	Introjected R.	5.86	0.79	5.94	0.61	0.861	5.98	0.54	5.84	0.61	0.216	0.270	0.498
	Identified R.	5.89	0.63	5.75	0.81	0.494	5.82	0.74	6.06	0.57	0.071	0.288	0.023 *
	IM to Accomplish	5.69	0.83	5.65	0.86	0.840	5.60	0.74	5.95	0.64	0.019 *	0.328	0.023 *
	IM to Experience	5.50	0.73	5.40	0.75	0.527	5.42	0.80	5.79	0.62	0.020 *	0.255	0.001 **
	IM to Know	5.53	0.95	5.60	0.86	0.597	5.47	0.83	5.92	0.57	0.006 **	0.367	0.007 **
	General IM	5.57	0.62	5.55	0.61	0.957	5.50	0.62	5.89	0.45	0.001 **	0.181	0.000 **
	SDI	7.99	2.91	7.68	2.83	0.393	7.56	3.08	9.50	1.75	0.001 **	0.067	0.000 **
Responsibility	Social Resp.	4.67	0.34	4.74	0.43	0.244	4.77	0.32	5.06	0.33	0.000 **	0.047 *	0.000 **
	Personal Resp.	4.71	0.55	4.78	0.51	0.384	4.75	0.47	4.99	0.39	0.005 **	0.774	0.002 **
BPN	Autonomy	3.33	0.60	3.52	0.63	0.174	3.41	0.57	3.95	0.48	0.000 **	0.123	0.000 **
	Competence	3.76	0.54	3.91	0.61	0.127	3.83	0.50	3.99	0.57	0.098	0.209	0.252
	Social Relatedness	3.89	0.73	3.93	0.61	0.997	3.82	0.61	4.16	0.60	0.010 *	0.285	0.035 *
	PMI	3.66	0.51	3.79	0.47	0.297	3.68	0.49	4.04	0.44	0.000 **	0.304	0.001 **
Social School Climate	School Climate	3.36	0.56	3.43	0.47	0.482	3.33	0.50	3.48	0.48	0.119	0.300	0.643
	Teacher Climate	3.66	0.47	3.60	0.48	0.388	3.58	0.41	3.82	0.60	0.049 *	0.227	0.045 *
IPA	IPA	2.79	0.84	2.87	0.79	0.506	2.70	0.97	3.34	0.76	0.000 **	0.014 *	0.000 **
Executive Functions	Verbal Fluency	9.52	2.02	9.82	1.77	0.416	9.76	1.62	10.30	1.49	0.110	0.065	0.003 **
	Planning	123.57	22.73	121.89	21.51	0.802	124.22	22.24	134.88	22.65	0.013 *	0.283	0.000 **

Note: * *p* < 0.05; ** *p* < 0.01; SD = Standard deviation; IM = intrinsic motivation; R = regulation; SDI = self-determination index; Resp. = responsibility; BPN = basic psychological needs; PMI = psychological mediator index; IPA = intention to be physically active.

## Data Availability

Not applicable.

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
