# Peer review of "The Effects of the ACTIVE VALUES Program on Psychosocial Aspects and Executive Functions"

_ijerph, 2022, doi:10.3390/ijerph20010595_

Round 1

Reviewer 1 Report

The text submitted for evaluation is an interesting contribution to find out the results of a physical education intervention proposal in various schools in the Autonomous Community of Murcia. The text analyzes the results of the intervention.

The text presents a correct scientific structure. It is a quantitative investigation, with an experimental group and a control group. The sample is composed of 102 students and 4 teachers.

The text presents discussion and conclusions.

In the scientific bibliography we find references up to 2022.

Author Response

Dear editor and reviewer:

Thank you very much for your feedback, we appreciate your positive view on the manuscript. The English language has been reviewed by an expert.

Regards.

Reviewer 2 Report

This manuscript presents a relevant topic to publish in IJERPH, which could be accepted with minor revisions. 

 The introduction provides adequate information and structure to set up the research questions raised in the manuscript; the methodology provides sufficient detail; the results section is sufficiently clear and precise; the discussion of results is based on previous literature.

After carefully reading your manuscript, I point out some aspects that must be improved and corrected:

    The authors should review the first sentence of the abstract. It must be coherent with the objective formulated in lines 137 -139. The words "active effects" it is not understood.

    Throughout the text, authors should try standardizing the following words: (programme; program) (psycho-logical; psychological). All are correct, but the authors must make a uniform option throughout the text.

    Some aspects of formatting should be corrected (spelling). Please, correct what is pointed out in the body of the manuscript;

    All statistical symbols must be in italics (N, n, p,  ....). 

     Moderate editing of the English language is required.

Author Response

We appreciate the comments and considerations to improve the manuscript. We specify the changes in the attached document. The English language has been reviewed by an expert.

Regards.

Reviewer 3 Report

Interesting title suggested worthy academic work so I am glad to assist in enhancing the quality of the presentation. 

Abstract informative and well-constructed. Too many key words, I do not think it is necessary to have them so many. 

Introduction - since your study is about the intervention program I would expect more examples of effective intervention designs mentioned in the introductory paragraphs - some of the interventions aimed at goal-setting strategies, other at combining the educational efforts within the school and family settings, some studies analyzed values equally important for physical education as Hellison's responsibility (like fair play in example, or in terms of motor development active BrainBreaks).  And except for Hellison's and et. al works you mostly included Spanish authors and Spanish origin papers (not mentioning the self-citations of your own works, which, even if some of them could be justified, add too much to the problem of references. It sets some limits to the presentation of your study. 

 I would also suggest re-structuring the Introduction section and when you deal with the examples of other interventions place it in one place of the Introduction body text, do not presented in separated paragraphs. 

Generally, Introduction needs to be re-worked. 

Methods section

It seems that selection of participants, although it was not random, but still  was well-defined. So were the research instruments, with one exception - you need to expand in more details on the Executive functions testing.

The whole intervention procedure was also described in details and permissions from all the legal agents (parents, Bioethical committee) were secured as well, which was good.

Results - I think the first line (p.7, lines317-319) should be moved to the Data analysis subsection, as this is not the part of the results in terms of the study objectives, but it is rather description of the statistical procedure. 

Discussion - I my opinion is too limited as well. I got the impression that it is a bit superficial, you are not tackling the problems in depths. Especially problematic for me is the claim that the Active Values program mediated changes in Executive functions - and you do not go on with explaining how those changes have been caused, what were the neurocognitive processes that initiated the change, how the teaching contents could bring about such changes at the level of functional operating nervous system? Here we come back to the testing of these functions - too little is known what and how was tested and why you have selected only those functions not other. In the Introduction you need to provide more rationale for your choices, and Discussion you should refer to more works on stimulation of EF in school-setting. 

Reference are very one-country narrowed. It does not look good, especially with your own works cited extensively.  

Author Response

(The authors gave the same response as above.)
